# Interplay between Podoplanin, CD44s and CD44v in Squamous Carcinoma Cells

**DOI:** 10.3390/cells9102200

**Published:** 2020-09-29

**Authors:** Lucía Montero-Montero, Jaime Renart, Andrés Ramírez, Carmen Ramos, Mariam Shamhood, Rocío Jarcovsky, Miguel Quintanilla, Ester Martín-Villar

**Affiliations:** 1Instituto de Investigaciones Biomédicas “Alberto Sols”, Consejo Superior de Investigaciones Científicas (CSIC)—Universidad Autónoma de Madrid (UAM), 28029 Madrid, Spain; luciamontero@iib.uam.es (L.M.-M.); jrenart47@gmail.com (J.R.); andres.ramirez1793@gmail.com (A.R.); carmenramosnebot@gmail.com (C.R.); ms.1bctb@gmail.com (M.S.); 2Departamento de Biotecnología—Instituto de Investigaciones Biosanitarias. Facultad de Ciencias Experimentales, Universidad Francisco de Vitoria (UFV), 28223 Madrid, Spain; rjarcovsky@iib.uam.es

**Keywords:** podoplanin, CD44s, CD44v, squamous carcinoma cells, protein–protein interaction

## Abstract

Podoplanin and CD44 are transmembrane glycoproteins involved in inflammation and cancer. In this paper, we report that podoplanin is coordinately expressed with the CD44 standard (CD44s) and variant (CD44v) isoforms in vivo—in hyperplastic skin after a pro-inflammatory stimulus with 12-*O*-tetradecanoylphorbol-13-acetate (TPA)—and in vitro—in cell lines representative of different stages of mouse-skin chemical carcinogenesis, as well as in human squamous carcinoma cell (SCC) lines. Moreover, we identify CD44v10 in the mouse-skin carcinogenesis model as the only CD44 variant isoform expressed in highly aggressive spindle carcinoma cell lines together with CD44s and podoplanin. We also characterized CD44v3-10, CD44v6-10 and CD44v8-10 as the major variant isoforms co-expressed with CD44s and podoplanin in human SCC cell lines. Immunofluorescence confocal microscopy experiments show that these CD44v isoforms colocalize with podoplanin at plasma membrane protrusions and cell–cell contacts of SCC cells, as previously reported for CD44s. Furthermore, CD44v isoforms colocalize with podoplanin in chemically induced mouse-skin SCCs in vivo. Co-immunoprecipitation experiments indicate that podoplanin physically binds to CD44v3-10, CD44v6-10 and CD44v8-10 isoforms, as well as to CD44s. Podoplanin–CD44 interaction is mediated by the transmembrane and cytosolic regions and is negatively modulated by glycosylation of the extracellular domain. These results point to a functional interplay of podoplanin with both CD44v and CD44s isoforms in SCCs and give insight into the regulation of the podoplanin–CD44 association.

## 1. Introduction

Squamous cell carcinoma (SCC) is one of the most frequent cancers worldwide. It arises in cells of stratified epithelia such as the epidermis, esophagus and mucosal linings of the upper airways, lips, mouth, salivary glands, throat and larynx (head and neck SCC, HNSCC). Cutaneous SCC is the second most common non-melanoma skin cancer—and although most of them are easily eradicated by surgical excision, a subset of these tumors become highly aggressive and show higher recurrence and metastatic rates, causing the death of patients [1]. On the other hand, HNSCCs had an incidence of more than 700,000 cases worldwide in 2018, with 42–56% mortality [2]. The main risk factors for cutaneous SCC are exposure to sunlight and immunosuppression. Smoking and excessive alcohol intake predispose to development of HNSCC. Currently, an increasing proportion of these tumors has arisen as a consequence of infection with human papilloma viruses [1,3]. Despite recent advances in the knowledge of the molecular landscape of SCCs [3], the factors contributing to the malignant behavior and clinical aggressiveness of many SCCs—particularly HNSCCs—remain poorly understood. In this respect, the emergence of novel biomarkers for these types of tumors, such as podoplanin and CD44 and the study of their functional interplay, may help to understand the biological mechanisms driving SCC invasion and metastasis.

Podoplanin (also called PA2.26, Aggrus, T1α, gp38 and D2–40) is a Type I transmembrane mucin-like sialoglycoprotein whose expression is upregulated in a variety of cancers, including SCCs [4,5,6,7] Podoplanin is also expressed in a wide range of normal tissues and cell types, such as lymphatic endothelial cells, mesothelial cells, glomerular podocytes (so its name), Type I alveolar cells, some type of neurons and subsets of fibroblasts and immune cells [6,8] Studies on podoplanin-deficient mice have shown that this glycoprotein plays a crucial role in the development of the lymphatic vasculature, heart and lungs. These mice display embryonic lethality due to cardiovascular problems [9,10] or die shortly after birth because of a respiratory failure [11] and exhibit a deficient lymphatic–blood vessel separation causing blood–lymphatic misconnections, lymphedema and blood-filled lymphatic vessels [12,13]. In adult tissues, podoplanin plays pivotal roles in lymphangiogenesis, platelet formation in the bone marrow and the immune response [4,8,14,15].

The podoplanin molecule lacks obvious enzymatic motifs, hence, it must exert its biologic and pathologic functions through protein–protein interactions. A number of partners, such as the C-type lectin-like receptor 2 (CLEC-2), ezrin and moesin members of the ERM (ezrin, radixin, moesin) protein family, CD9 tetraspanin, and the standard isoform of the hyaluronan receptor CD44s, among others, have been found to interact with podoplanin in different cell types and contexts [4]. Podoplanin binds to ERM proteins through a juxtamembrane cluster of basic amino acids in its cytoplasmic (CT) domain, allowing the anchorage of the glycoprotein to the actin cytoskeleton and formation of cell-surface protrusions, such as filopodia and ruffles related to cell migration [16,17,18]. Podoplanin-ERM interaction is critical for activation of small Rho GTPases and induction of epithelial–mesenchymal transitions (EMTs) during embryogenesis [10] and malignant progression [17,19]. In SCC cells, podoplanin–ERM interaction is also critical for the stable localization of podoplanin at invadopodia, specialized cell-surface protrusions involved in tumor invasiveness [20]. On the other hand, in fibroblast reticular cells (FRCs), podoplanin controls cell stiffness by inducing actomyosin contractility via ERM binding and RhoA GTPase activation [14,21]. CLEC-2 is expressed in platelets and some immune cells, and podoplanin binds CLEC-2 through four platelet aggregation-stimulating (PLAG) motifs located in tandem in its extracellular (EC) domain. Glycosylation of Thr-52 and Thr-85 in PLAG-3 and PLAG-4 motifs, respectively, appear to be critical for podoplanin–CLEC–2 binding [22,23,24]. Upon binding to platelet CLEC-2, podoplanin in lymphatic endothelial cells (LECs) induces platelet aggregation and activation allowing the correct separation of the lymphatic and blood vasculatures during embryogenesis [8,25,26]. Similarly, the interaction of podoplanin in FRCs to CLEC-2 in megakaryocytes is involved in platelet production in the bone marrow [27], and binding of FRCs to migratory dendritic cells via podoplanin–CLEC–2 interaction counteracts podoplanin-induced actomyosin contractility, allowing FRC spreading and lymph node expansion upon initiation of an immune response [14,21]. Podoplanin expressed in tumor cells facilitate hematogenous metastasis by binding to platelet CLEC-2. This interaction induces platelet activation and aggregation all around tumor cells, protecting them from shear stress and the immune system attack [28,29,30]. Interestingly, binding of CD9 to podoplanin in tumor cells antagonizes podoplanin–CLEC–2 interaction, inhibiting platelet aggregation and metastasis [29].

CD44 is a highly polymorphic Type I transmembrane glycoprotein which is ubiquitously expressed in the body [31]. It is the main receptor for hyaluronic acid, an abundant component of the extracellular matrix, and it also binds other extracellular ligands such as osteopontin, collagens and matrix metalloproteases [32]. CD44 is encoded by a single gene but gives rise to a wide array of isoforms by alternative splicing. Thus, in humans, the standard isoform CD44s is encoded by ten constant exons, while the variant isoforms (CD44v) include nine possible variant exons in different combinations, in addition to the ten constant exons that enlarge the extracellular domain [33,34]. CD44 is involved in cell adhesion, migration and a range of pathophysiological processes as inflammation, wound healing and cancer [35,36]. CD44s and CD44v isoforms are overexpressed in different types of cancers, and particularly CD44v isoforms are cancer stem cell (CSC) markers that have been associated with self-renewal, tumor initiation, metastasis and resistance to chemo- and radiotherapy in different types of cancer, including HNSCCs [37,38,39]. In a previous report, we found that podoplanin binds CD44s on cell-surface protrusions associated with motility of SCC cells and that both glycoproteins cooperate to promote directional migration of SCC cells [40].

In this article, we have characterized the expression of CD44v isoforms in human SCC cell lines and report the coordinate expression of podoplanin with CD44s and CD44v isoforms in vivo and in vitro. Colocalization and co-immunoprecipitation experiments suggest that podoplanin interacts with both CD44s and CD44v isoforms in SCC cells through the transmembrane (TM) domain. The podoplanin–CD44 interaction is modulated positively by the CT tail and negatively by glycosylation of the EC domain.

## 2. Materials and Methods

### 2.1. Skin TPA Treatment and Chemical Carcinogenesis

All the experimental procedures with mice were approved by the internal ethical research and animal welfare committee (IIBM, CSIC) and by the local authorities (Comunidad de Madrid, PROEX 37/14). They complied with the European Union (Directive 2010/63/UE) and Spanish Government guidelines (Real Decreto 53/20133). Seven to nine weeks old C57Bl/6 mice were shaved 24 h before the initiation of the treatment. For skin TPA treatment, 200 μL of 62.5 μg/mL TPA solution in acetone was applied to the shaved skin of each animal. Control animals were treated with the same volume of acetone. Animals were killed at 12, 24, 48, 72, 96, 120 and 144 h posttreatment and the skin was harvested, divided in two pieces and immediately frozen or fixed in 10% formaldehyde and embedded in paraffin for histological analysis.

Skin tumors were induced in FVB/N mice by initiation with 32 µg of DMBA in acetone and promotion once per week for 14 weeks with 62.5 μg/mL TPA, as previously described [41]. A total of 10 six-week-old mice were used. Animals bearing tumors were sacrificed at 12, 15, 20, 29 and 30 weeks post-DMBA initiation. Tumors were excised, divided into two regions and immediately frozen or fixed in 10% formaldehyde and embedded in paraffin. Tumors were histologically typed by hematoxylin–eosin staining of paraffin sections and by expression analyses of several differentiation/progression markers. Tumors shown in this work were classified as moderately differentiated SCCs.

### 2.2. Cell Culture

The origin, characteristics and cell culture conditions of murine epidermal cell lines and human SCC cell lines are described in Appendix A, respectively. Human SCC cell lines were short tandem repeat (STR) profiled and their genotypes compared with that of the database (ATCC STR database or Zhao and coworkers [42]). All cell lines were maintained at 37 °C in a 5% CO_2_ humidified atmosphere.

### 2.3. Plasmids and Cell Transfections

CD44s and CD44v constructs tagged with enhanced green fluorescent protein (eGFP) or hemagglutinin (Ha) were obtained by PCR amplification using primers that carry convenient restriction sites to facilitate subcloning into pEGFP-N1 or pcDNA3-Ha vectors, as described in Appendix A. mCherry- and FLAG-tagged wild-type and mutant podoplanin constructs PDPN-ΔEC, PDPN-ΔCT, PDPN-TMCD45, PODPN-G137L and PDPN-QN.N were subcloned from cDNAs fused to eGFP [17] using Phusion high fidelity DNA polymerase (New England Biolabs) and specific primers, which are described in Appendix A. Podoplanin mutant constructs PDPN-ΔECQN.N, PDPN-S/Tm, PDPN-ΔPLAG3, PDPN-PLAG3m, PDPN-PLAG3Tm, PDPN-PLAG4Tm, PDPN-I1, PDPN-I2, PDPN-TMSYN, PDPN-TMGPA and PDPN-TMERBB2 were obtained by directed mutagenesis of wild-type podoplanin sequence [43], using primers described in Appendix A. In PDPN-S/Tm, the following Ser and Thr residues were mutated to Ala: Thr25, Thr32, Thr34, Thr35, Thr52, Thr55, Thr65, Thr66, Thr85, Ser86, Ser88, Thr89, Thr117, Thr119 and Thr120.

Transient transfections in HN5 and HEK-293T cells were performed with 1 μg of cDNAs, using either FuGENE 6 (Promega Biotech Ibérica, Madrid) or Lipofectamine 2000 (Invitrogen, Thermo Fisher Scientific, Madrid, Spain), following the manufacturer indications. After 24 h, cells were either lysed for immunoprecipitation and Western blot experiments or seeded on glass coverslips for immunofluorescence analysis.

### 2.4. RT–PCR

Reverse transcription was performed as previously described [17]. The primers used to amplify transcripts of all CD44 isoforms (hs5/hs3) and primers for exon-specific RT–PCR analysis [44] are described in Appendix A. PCR products were obtained after 35 cycles of amplification with annealing temperatures of 57–60 °C.

### 2.5. Immunofluorescence and Confocal Microscopy

HN5 cells co-transfected with PDPN-mCherry and CD44 isoforms fused to eGFP were seeded on glass coverslips, fixed with 3.7% formaldehyde for 20 min and permeabilized with 0.05% Triton X-100/1x PBS for 10 min. Confocal images were acquired on a spectral LSM/10 laser-scanning microscope (Zeiss), using objectives Plan-APOCHROMAT 40× and 63×. Images were assembled using the FIJI program.

Double-label immunofluorescence detection of podoplanin and CD44v isoforms in mouse-skin tumors was performed in deparaffinized sections, after antigen retrieval with 1 mM EDTA, using a rat monoclonal antibody (mAb PA2.26; 1:100) specifically recognizing mouse podoplanin [45] and rabbit polyclonal Abs (1:200) recognizing CD44v3 and v6 exons (Millipore, Merck Chemical Life Science SA, Madrid, Spain). Alexa Fluor 488 goat anti-rat IgG and Alexa Fluor 546 goat anti-rabbit IgG (Molecular Probes) were used as secondary Abs. Nuclei were stained with a 1 μg/mL solution of 4′,6-diamino-2-phenilindole (DAPI; Sigma-Aldrich, Merck Chemical Life Science SA, Madrid).

### 2.6. Western Blot and Co-Immunoprecipitation Experiments

For detection of podoplanin and CD44 in Western blots, cells or tissues were lysed in buffer RIPA (0.1% SDS, 0.5% sodium deoxycholate, 1% Nonidet *P*-40, 150 mM NaCl, 50 mM Tris-HCl pH 8.0) and a cocktail of protease inhibitors (Calbiochem) and phosphatase inhibitor (500 mM NaF). Samples containing the same amount of protein (20–30 μg) were run on 10% SDS-PAGE and transferred to Immobilon-P PVDF membranes (Millipore, Bedford, MA, USA). Filters were then immunoblotted with the following Abs: anti-Ha (Merck), anti-mouse podoplanin (PA2.26 mAb [45]), anti-human podoplanin (NZ1 mAb purchased from Acris Antibodies), rat mAb IM7 against CD44 ectodomain (generously provided by Dr Helen Yarwood; The Institute of Cancer Research, London, UK) and mouse mAbs against α-tubulin, β-actin (Merck) and GAPDH (Millipore).

For co-immunoprecipitation experiments, HEK293T cells were transiently co-transfected with expression vectors as indicated above. After 24 h, cells were lysed in IP buffer (50 mM Tris-HCl pH 7.5, 100 mM NaCl, 5 mM EDTA, 0.5% NP-40, 10% glycerol) supplemented with a cocktail of protease and phosphatase inhibitors. Tagged podoplanin-FLAG protein was immunoprecipitated from about 750 μg of cell lysate using anti-FLAG (Merck Chemical Life Science SA, Madrid) mAb and protein G Dynabeads (Thermo Fisher Scientific, Madrid). After incubation overnight at 4 °C, beads were washed 4 times with lysis buffer and eluted in Laemmli buffer. The co-immunoprecipitated products were detected in a Western blot using the rabbit anti-Ha Ab (Merck).

## 3. Results

### 3.1. The Expression of Podoplanin and CD44 Is Coordinately Induced by TPA in Mouse Skin

We have previously reported that podoplanin expression is induced in mouse-skin upon a pro-inflammatory stimulus with 12-*O*-tetradecanoylphorbol-13-acetate (TPA) [45]. In order to ascertain whether TPA was able to induce CD44 together with podoplanin in vivo, the skin of mice was treated with TPA and the expression of podoplanin and CD44 analyzed by Western blotting at different times posttreatment. As expected, TPA induced a skin hyperplasia involving swelling of the epidermis and recruitment of inflammatory cells into the dermis. These effects were mainly observed between 24 h and 72 h posttreatment (Figure 1A), which coincided with maximal levels of podoplanin and CD44 proteins, both of which were coordinately induced by TPA (Figure 1B). After 144 h posttreatment, the expression of podoplanin and CD44 dropped to roughly basal levels and the skin recovered its normal appearance (Figure 1A,B). It is worth to mention that expression of both CD44s and CD44v were uniformly enhanced by TPA in the skin together with podoplanin.

### 3.2. Podoplanin Co-Localizes with CD44v in Chemically-Induced Skin Tumors In Vivo and Is Co-Expressed with CD44s and CD44v in Transformed Mouse Epidermal Cell Lines

A coordinate upregulation of podoplanin with CD44s and CD44v is also observed in vivo during malignant progression of mouse-skin chemical carcinogenesis [40]. Yet, whereas a link between podoplanin and CD44s associated with cell migration was found in SCCs [40], the possible cooperation of podoplanin with CD44v was not investigated. Since CD44v isoforms containing exons v3 and v6 were found to be expressed in SCCs [37], we analyzed the potential colocalization of podoplanin with CD44v isoforms in well differentiated mouse-skin SCCs using anti-podoplanin, anti-v3 and anti-v6 specific Abs. As shown in Figure 2, a clear colocalization of podoplanin with CD44v isoforms containing v3 and v6 was observed.

We also analyzed the expression of podoplanin and CD44 in cell lines representative of different stages of mouse-skin carcinogenesis: immortalized non-tumorigenic keratinocytes (MCA3D), benign papilloma (PB), well to moderately differentiated SCCs (PDV, B9) and poorly differentiated spindle cell carcinomas (SpCCs; A5, CarC and CarB). The profile of RT–PCR expression showed a switch from CD44v transcripts (500–900 pb) to CD44s transcripts (~300 pb) in SpCC cells when compared with the other cell lines (Figure 3A). This CD44v–CD44s switch correlated with increased podoplanin expression, as previously reported [40]. In addition, SpCC cells, but not the other cell lines, expressed low levels of a PCR fragment of ~380 pb (Figure 3A, asterisk), which after isolation and sequencing was identified as a CD44v isoform containing v10 as the only variant exon (CD44v10). Podoplanin was expressed in all cell lines, except in MCA3D. Western-blot analysis showed that while podoplanin was undetectable from non-tumorigenic MCA3D and papilloma PB cells, it was co-expressed with CD44v isoforms of 100–180 kDa in SCC cell lines PDV and B9. Considering the sizes of CD44v transcripts and proteins, it is likely that CD44v isoforms expressed in these murine SCC cells are similar to those expressed in human SCC cell lines, i.e., CD44v3-10, CD44v6-10 or CD44v8-10 (see below). In SpCC cells A5, CarC and CarB, a faint band above CD44s, likely corresponding to CD44v10, was detected in the blot (Figure 3B, asterisk). Likewise, in these cells, podoplanin is co-expressed with CD44v10 and CD44s.

### 3.3. Characterization of CD44v Isoforms Expressed in Human SCC Cell Lines

Next, we studied the expression of podoplanin and CD44 by RT–PCR in a panel of human SCC cell lines from different origins: pharynx (HN30, Fadu), tongue (HN5), submandibular (A253), facial epidermis (SCC13) and non-tumorigenic immortalized skin keratinocytes HaCaT. We used first the pair of primers hs5/hs3 aimed to amplify all CD44 transcripts at once (Figure 4A). All cell lines co-expressed CD44s (~400 bp) with transcripts corresponding to different CD44v isoforms (800–1500 bp) and podoplanin (~500 bp), except HaCaT, in which podoplanin mRNA was undetectable (Figure 4B). The pattern of CD44v mRNA expression in these cell lines was more heterogeneous and more variable than that seen in mouse epidermal cell lines (Figure 3A). Consequently, in order to characterize the specific CD44v isoforms expressed, we selected three representative human cell lines: HaCaT, HN5 and A253. These cell lines express three main CD44v transcripts of ~1500, ~1013 and ~800 bp (denoted as 1, 2 and 3, respectively) besides a transcript of ~400 bp corresponding to CD44s (Figure 4C). To determine the exon composition in the CD44v transcripts expressed in these cell lines, we carried out exon-specific RT–PCR analysis [44]. To this end, a primer of the 3′ constant region (hs3) and one of the variant exon-specific 5′ primers (v2–v10) were used (Figure 4A). As shown in Figure 4E, a typical ladder-like pattern was observed in all cases, indicating that all of variant exons are expressed in the three cell lines. Therefore, the larger CD44v transcript 1 of ~1500 bp should correspond to CD44v2-10 and taking into account the relative sizes of transcripts 2 and 3, they may correspond to CD44v6-10 and CD44v8-10. The isolation and sequencing of transcripts 1, 2 and 3 confirmed our predictions, except that transcript 1 was CD44v3-10 instead of CD44v2-10. Therefore, it seems that CD44v2-10 is not clearly expressed in these human SCC cell lines. By using the pair of primers C5/C9 (Figure 4A) in our RT–PCR analysis, we also identified the presence of shorter CD44v and CD44s transcripts (below 800–1500 bp and below 400 bp) in HN5 cells that were denoted as CD44vC9 and CD44sC9, respectively (Figure 4D). These transcripts correspond to CD44v and CD44s isoforms containing truncated cytosolic domains [33].

Western-blot analysis revealed the presence of at least five polypeptide bands in the cell lines (Figure 4F). The bands of lowest (~85 kDa) and largest (~250 kDa) sizes likely correspond to CD44s and CD44v3-10, respectively, while those of ~150 kDa and ~130 kDa are consistent with CD44v6-10 and CD44v8-10, respectively, as demonstrated below (Figure 6). A fifth band of ~180 kDa detected in some, but not all the cell lines could represent CD44 isoform, either CD44s or CD44v, additionally modified by glycosylation. Overall, CD44v and CD44s protein expression levels differed widely, both quantitatively and qualitatively, among the cell lines (Figure 4F). Podoplanin was expressed in all SCC cell lines albeit at highly variable levels (Figure 4G). Nevertheless, a rough correlation between the levels of CD44 and podoplanin could be observed in the human SCC cell lines (Figure 4, compare panels F and G). The different molecular masses of podoplanin proteins observed in the cell lines are due to different degrees of glycosylation as previously demonstrated [18,46].

### 3.4. Podoplanin Colocalizes with CD44s and CD44v Isoforms at the Plasma Membrane of Human SCC Cells

We have previously reported that CD44s and podoplanin colocalize at the plasma membrane of HN5 cells [40]. Therefore, we analyzed in this work whether podoplanin colocalizes with CD44v isoforms co-expressed in SCC cells. For this purpose, we engineered constructs encoding CD44v3-10, CD44v6-10 and CD44v8-10 fused to eGFP that were co-transfected with a construct encoding podoplanin fused to mCherry in HN5 cells. The truncated CD44sC9 isoform fused to eGFP was included in this analysis to ascertain whether a shorter CD44s cytoplasmic tail may affect its colocalization with podoplanin. As shown in Figure 5, all CD44v and CD44sC9 isoforms colocalized with podoplanin at the plasma membrane, concentrated at cell-surface protrusions and cell–cell contacts. These results suggested that the physical association between podoplanin and CD44 may not be exclusive of the CD44s isoform, but it also may include CD44v and CD44sC9 isoforms.

### 3.5. Podoplanin Interacts with CD44s and CD44v Isoforms

Consequently, we studied the interaction of CD44v3-10, CD44v6-10, CD44v8-10, CD44s and CD44sC9 with podoplanin by co-immunoprecipitation analysis. To this end, we engineered constructs encoding CD44 isoforms tagged with Ha epitope at the C-terminus. Western-blot analysis of these constructs expressed in HEK293T cells confirmed the expected molecular sizes for CD44 isoforms (Figure 6A); compare with the endogenous CD44v expression shown in (Figure 4F). The bands of lower size seen in all lanes correspond to less glycosylated forms as demonstrated previously [40]. Next, HEK293T were co-transfected with these CD44 constructs and a plasmid encoding podoplanin tagged with FLAG. Immunoprecipitation experiments confirmed that CD44s co-precipitated with podoplanin [40] and revealed that CD44v isoforms also bind podoplanin (Figure 6B). Whereas the anti-FLAG antibody pulled down CD44s and CD44v6-10 with similar efficiencies (Figure 6B, lanes 4 and 6, respectively) the efficiency of co-immunoprecipitation was lower for CD44v8-10 (Figure 6B, lane 7), and, particularly, for CD44v3-10 (Figure 6B, lane 5). These differences could be due to the consistently lower level of expression raised by CD44v3-10 or CD44v8-10 after transfection (see Figure 6B, input). On the other hand, although the level of expression raised by CD44sC9 was similar to that of CD44v3-10 (Figure 6B, input), no band could be seen in the precipitate unless a longer exposure of the blot was performed (Figure 6B, lanes 8 and 8L, respectively), pointing to an involvement of the CD44 cytoplasmic tail in the interaction with podoplanin. Interestingly, co-precipitated CD44 isoforms pulled down by the anti-FLAG antibody had lower molecular masses than those of fully glycosylated mature forms (Figure 6B, compare IP and input panels). This observation was reported before [40] and can be explained by the fact that glycosylation of the extracellular domain of podoplanin negatively modulate podoplanin–CD44 interaction (see below).

### 3.6. Analysis of the Structural Domains Involved in Podoplanin–CD44 Interaction

To analyze the regions of the podoplanin molecule involved in the interaction with CD44, we performed immunoprecipitation experiments with HEK293T cells co-expressing CD44s-Ha and several constructs encoding podoplanin mutants tagged with FLAG (Figure 7A). These mutants included podoplanin proteins lacking the cytoplasmic domain (PDPN-ΔCT), the extracellular domain that was substituted by eGFP (PDPN-ΔEC), the ERM-binding site (PDPN-QN.N) and the CLEC-2-binding site (PDPN-ΔPLAG3 and PDPN-PLAG3m). We also used a podoplanin construct with stochastic mutations in potential O-glycosylation Ser and Thr residues of the EC domain (PDPN-S/Tm), as well as transmembrane mutant proteins harboring a heterologous TM domain (PDPN-TMCD45) or a mutated GXXXG TM motif involved in podoplanin self-assembly and its association with lipid rafts (PDPN-G137L). PDPN-ΔCT, PDPN-ΔEC, PDPN-QN.N, PDPN-TMCD45 and PDPN-G137L mutants have been previously characterized in our laboratory [17,49], while PDPN-ΔPLAG3 and PDPN-PLAG3m mutants were characterized by others [22,50]. As shown in Figure 7B, wild-type podoplanin was able to bind incompletely glycosylated CD44s forms (Figure 7B, lane 2), but deletion of the EC domain allowed podoplanin to interact with fully glycosylated CD44s forms (Figure 7B, lane 3). Mutation of the ERM-binding site neither affect interaction of wild-Type PDPN with CD44 (Figure 7B, lane 11) nor the interaction of mutant PDPN-ΔEC (Figure 7B, lane 4). However, deletion of the whole CT domain prevented PDPN–CD44 binding (Figure 7B, lane 10). Mutation of several amino acids of the EC PLAG3 domain (EDDVVTPG to AADVVAPG), including a Thr52 whose glycosylation is crucial for podoplanin binding to CLEC-2 [50], did not affect PDPN–CD44 interaction (Figure 7B, lane 6). Nonetheless, deletion of the whole PLAG3 motif favored interaction with highly glycosylated CD44s forms (Figure 7B, lane 7), albeit not as much as deleting the entire EC domain (Figure 7B, lane 3). Neither substitution of the TM domain by that of CD45 nor mutation of the GXXXG motif (GXXXL) involved in podoplanin homodimerization and incorporation into lipid rafts [49] did impair PDPN–CD44 binding, instead it was slightly enhanced (Figure 7B, lanes 8 and 9, respectively). Finally, the stochastic mutations of potential O-glycosylation Ser and Thr residues (see Materials and methods) within the EC domain did not affect PDPN–CD44 interaction (Figure 7B, lane 5). Taken together, these results suggest the involvement of the CT domain in podoplanin–CD44 interaction that is not mediated by its binding to ERM proteins, as previously suggested [40]. On the other hand, podoplanin–CD44 interaction appears to be negatively modulated by glycosylation of the podoplanin EC domain.

Because the mutant PDPN-S/Tm had no effect in the binding of podoplanin to CD44, we reasoned that glycosylated Ser and Thr impairing podoplanin–CD44 interaction should be clustered into specific regions of the EC domain. Based in the available data from the literature [22,24,50,52], we defined four EC regions previously found experimentally to be glycosylated that could influence podoplanin–CD44 interaction (Figure 8A). These regions are Thr52 and Thr85 located within the PLAG3 and PLAG4 motifs, respectively and two zones located between the PLAG3 and PLAG4 motifs (named island 1, comprising amino acids 65–76) and downstream PLAG4 (named island 2, comprising amino acids 98–110). Islands 1 and 2 contain several glycosylated Ser and Thr residues that were mutated to Ala, giving rise to the following mutants: PDPN-I1, PDPN-I2 and PDPN-I1/2 (the combination of the two first). Similarly, Thr85 and Thr52 were also mutated to Ala, giving rise to PDPN-PLAG4Tm, PDPN-PLAG3Tm and PDPN-PLAG3/4Tm mutants (Figure 8A). PDPN-PLAG3Tm, PDPN-PLAG4Tm and PDPN-PLAG3/4Tm mutants co-precipitated CD44s with the same efficiency as PDPN wild-type (Figure 8C, lanes 2–5), as previously found with PDPN-PLAG3m (Figure 7B, lane 6), indicating that glycosylation of Thr52 and Thr85 did not affect podoplanin–CD44 binding. On the contrary, mutant PDPN-I1m and, to less extent, PDPN-I2m favored co-precipitation with CD44 and pulled down higher glycosylated CD44 forms (Figure 8C, lanes 6 and 7, respectively), indicating that glycosylation of Ser and Thr residues located in islands 1 and 2 lessen podoplanin–CD44 interaction, a fact confirmed with the PDPN-I1/2m combined mutant, in which mutations in both regions acted synergistically to favor podoplanin–CD44 binding (Figure 8C, lane 8).

We also investigated the role of the TM domain by using other heterologous transmembrane podoplanin mutants in addition to PDPN-TMCD45: PDPN-TMSYN, PDPN-TMGPA and PDPN-TMERBB2, in which the TM domain of podoplanin was substituted by that of synaptobrevin, glycophorin A and the tyrosine kinase receptor ERBB2 (also known as HER2 or Neu), respectively (Figure 8B). The co-immunoprecipitation of podoplanin with CD44 was not affected by the TM region of ERBB2 (Figure 8D, lanes 2 and 5), while the TM domains of synaptobrevin and glycophorin A prevented podoplanin–CD44 binding (Figure 8D, lanes 3 and 4, respectively).

In summary, this extensive immunoprecipitation analysis suggests a main role of the TM and CT domains in podoplanin–CD44 interaction, which is hampered by glycosylation of the EC domains.

## 4. Discussion

In a previous report, we demonstrated the physical cooperation of podoplanin and CD44s in promoting directional migration of SCC cells [40]. In the present work, we further characterize this interaction biochemically, providing an important insight into its regulation. More significantly, we report that podoplanin can also bind to some variant forms of CD44 and demonstrate that podoplanin is co-expressed, not only with CD44s, but also with CD44v in vivo, during skin remodeling and SCC development and in vitro, in mouse and human SCC cell lines.

Podoplanin expression is coordinately upregulated with both CD44v and CD44s isoforms in the skin upon a proinflammatory stimulus with TPA, and the expression of these proteins is coincidentally associated with a skin hyperplasia. In addition to stimulation with TPA, the expression of podoplanin is induced in basal epidermal keratinocytes and dermal fibroblast-like cells during wound healing and psoriasis [45,53,54,55]. The induction of podoplanin in keratinocytes under these conditions is more likely associated with cell migration, since the knockdown of podoplanin in primary human keratinocytes increases E-cadherin expression and inhibits cell motility [55], while its overexpression stimulates cell migration and induces EMT [16,17]. The pro-migratory function of podoplanin in epidermal keratinocytes may be exerted in association with CD44, as found by us in SCC cells [40]. Shatirishvili and coworkers have found that the conditional knockout of CD44 in basal epidermal keratinocytes delay wound healing and reduces epidermal hyperplasia upon stimulation with TPA [56]. Whether these defects were due to the lack of CD44s, CD44v or both was not investigated. Consequently, it would be interesting to analyze the functional involvement of podoplanin association with both CD44s and CD44v in keratinocytes during epidermal remodeling and regeneration.

A switch in CD44 isoform expression from CD44v to CD44s has been associated with EMT of epithelial cell lines [57,58]. This change also occurs in keratinocyte cell lines representative of different stages of mouse-skin carcinogenesis during progression to highly aggressive SpCCs that have lost the epithelial phenotype and acquired fibroblastic features ([40]; this paper). In addition to the switch CD44v–CD44s, we report here the induction of CD44v10 as a novel attribute of SpCC cell lines. It remains to be investigated whether CD44v10 expression is also induced in SpCCs in vivo and whether it is linked to EMT. Interestingly, Ma and coworkers found that low levels of CD44v10 together with high levels of CD44s expression was a characteristic of stromal fibroblasts from cows and humans [59]. Expression of CD44 isoforms containing exon v10 is associated with metastasis in lymph nodes, resistance to radiotherapy and shorter disease-free survival in HNSCCs [60], as well as in other malignancies of hematopoietic origin, such as cutaneous leukemias and lymphomas [61]. Specifically, CD44v10 is expressed in Hodgkin’s lymphomas correlating with tumor recurrence and poor prognosis [62].

In this work, we also characterized CD44v3-10, CD44v6-10 and CD44v8-10 as the main CD44v isoforms co-expressed with CD44s in human SCC cell lines. In addition, truncated CD44v and CD44s isoforms with shorter cytoplasmic tails (CD44vC9 and CD44sC9) were found to be expressed in SCC cells. These truncated CD44 isoforms have been described in the literature [33], but their biologic relevance is still unknown. These results indicate the presence of an enormous variety of CD44 isoforms in carcinoma cells that may fulfil overlapping and distinct functions, hindering the studies on the role of CD44 in cancer. Podoplanin was found to colocalize with CD44s and CD44v isoforms at plasma membrane protrusions related to cell motility, such as filopodia and ruffles and in cell–cell contacts. Colocalization of podoplanin with CD44v6 and CD44v3 isoforms was also found in vivo in chemically induced skin SCCs. Co-immunoprecipitation experiments suggest that CD44s and CD44v isoforms are able to bind podoplanin, although with distinct affinities: CD44s = CD44v6-10 > CD44v8-10 > CD44v3-10 >>> CD44sC9. The bases for these differences are presently unknown. Nevertheless, these data suggest a functional interplay of podoplanin with CD44v isoforms in SCCs, as previously found with CD44s [40]. The expression of CD44v3-10, CD44v6-10 and CD44v8-10 has been correlated with malignant progression and chemo-resistance in different types of cancer, including SCCs [37,38,39]. Moreover, some of these CD44v isoforms have been related to stemness in SCC cells. Thus, elevated levels of CD44v8-10 in esophageal SCC patients treated with chemoradiotherapy were associated with poorer prognosis and the presence of this isoform in CSCs [63]. In addition, expression of CD44v3 isoforms has been linked to stemness and chemoresistance in HNSCCs [64,65]. Since podoplanin expression has been correlated with poor prognosis in skin, esophageal and HNSCCs [6], it is tempting to speculate that podoplanin may cooperate with CD44v isoforms to promote malignant progression. On the other hand, podoplanin has been proposed as a CSC marker in esophageal and cervical SCCs [66,67,68,69]—although its relationship with cancer stemness is far to be clear [4]. Therefore, further studies need to be done in order to unveil the functional role of podoplanin–CD44v association in CSCs and tumor progression.

The primary structural region involved in podoplanin–CD44 interaction seems to be the TM domain, since substitution of this podoplanin region by that of synaptobrevin or glycophorin A largely inhibited or completely abolished this interaction, while heterologous podoplanin molecules with TM regions from CD45 and ERBB2 bound CD44 with similar efficiencies as wild-type podoplanin. These results strongly suggest that there are structural motifs within the TM domain of podoplanin shared by ERBB2 and CD45—but not by synaptobrevin and glycophorin A, which are involved in podoplanin–CD44 interaction. In support of this, ERBB2 has been found to bind CD44 [70,71], although the structural regions involved in this binding were not investigated. It is possible that ERBB2–CD44 interaction occur through the TM domains. With respect to the leukocyte tyrosine phosphatase CD45, as far as we know there is no report describing its binding to CD44. Currently, we are investigating the structural and conformational TM requirements for podoplanin–CD44 interaction by comparative analysis of the TM sequences of all these proteins. Among the TM motifs important for podoplanin–CD44 interaction, the GXXXG region involved in podoplanin self-assembly and association with lipid rafts can be discarded, since the mutant PDPN-G137L, unable to fulfil these activities [49], bind efficiently CD44. Considering that CD44 is also a lipid raft protein [72], this fact allows podoplanin–CD44 interaction to occur within lipid rafts otherwise the recruitment of podoplanin to these membrane platforms could be compromised. CD44 is not the only partner known to interact with podoplanin through the TM region, since tetraspanin CD9, which has four TM domains, was reported to bind podoplanin through TM domains 1 and 2 in fibrosarcoma cells [29].

In addition to the TM domain, the CT tails appear to be relevant for podoplanin–CD44 interaction. This is based in the following observations: first, the podoplanin binding efficiency of CD44sC9, containing a truncated cytoplasmic domain, was highly decreased with respect to the full-length isoform CD44s; and second, deletion of the podoplanin CT tail (PDPN-ΔCT mutant) largely inhibited podoplanin–CD44 interaction. Members of the ERM protein family bind a cluster of basic amino acids in the CT tails of both podoplanin [16,17] and CD44 [73], linking these glycoproteins to the actin cytoskeleton. Nonetheless, podoplanin–CD44 interaction appears not to be mediated by ERM proteins, since mutant PDPN-QN.N unable to bind ERM proteins [17] efficiently bound CD44 [40]. On the other hand, deletion of the EC domain (PDPN-ΔEC mutant) not only did not impair the interaction with CD44 but allowed to bind highly glycosylated mature CD44 forms. This finding was demonstrated for CD44s (Figure 7B), as well as for CD44v isoforms (not shown) and is in contrast with our previous report in which a construct PDPN-ΔEC was unable to co-immunoprecipitate CD44 [40]. This discrepancy is likely due to the different strategies and PDPN-ΔEC constructs used in the co-immunoprecipitation experiments. The construct used in our previous report encoded a podoplanin mutant in which the EC region was substituted by eGFP, similar to the one used here, but without the FLAG tag and immunoprecipitation was performed with an anti-GFP Ab; thus, the binding of the Ab to the N-terminal region of the molecule could have affected the interaction with CD44. The results obtained with the current PDPN-ΔEC construct and mutants PDPN-I1m, PDPN-I2m and PDPN-I1/2m suggest that glycosylation of the EC domain negatively modulates podoplanin–CD44 interaction, likely because of the repulsive forces provided by the negative charges of sialic acid present in O-glycans [16,18]. The O-glycosylated Ser and Thr modulating podoplanin–CD44 interaction are clustered in the so-called islands 1 and 2 located between the PLAG3 and PLAG4 motifs and downstream of PLAG4, respectively. In contrast, Thr52 and Thr 85 within the PLAG3 and PLAG4 motifs, respectively, that are implicated in podoplanin–CLEC–2 interaction [24,50] did not affect podoplanin–CD44 binding. In this respect, it has been suggested a functional coordination between podoplanin binding to CLEC-2 and binding to CD44/CD9 in FRCs during the immune response. In FRCs, podoplanin promotes actomyosin contractility through its binding to ERM proteins and activation of RhoA GTPase, and binding of CLEC-2 located on migratory dendritic cells inhibits podoplanin-induced contractility, resulting in FRC spreading and elongation to allow a rapid lymph node expansion [21,74]. In a recent report, de Winde and colleagues suggest that upon binding of CLEC-2 to FRCs, podoplanin is recruited to cholesterol-rich domains (lipid rafts) where interacts with membrane partners CD44 and CD9 to form cell protrusions and to spread [75]. In fibrosarcoma cells, podoplanin–CD9 interaction neutralizes podoplanin-mediated platelet aggregation via binding to CLEC-2 and suppresses metastasis [29]. It would be interesting to investigate the interplay between the different podoplanin binding partners CD44, CD9 and CLEC-2 in SCC cells.

In summary, here we present evidence of a physical interaction of podoplanin with CD44s and CD44v isoforms expressed in human SCC cells, suggesting a functional interplay between podoplanin and CD44 likely associated with cancer cell invasion. A background for this assumption is our demonstration that podoplanin cooperates with CD44s in SCC cells to promote directional migration [40]. Since both podoplanin and CD44 have been described as components of tumor invasion-related invadopodia [20,76,77,78,79], we are currently investigating the role of podoplanin interaction with CD44s and CD44v in invadopodia formation and activity.

## 5. Conclusions

In the present study, we report that podoplanin interacts with both CD44s and CD44v (CD44v3-10, CD44v6-10 and CD44v8-10) isoforms expressed in SCC cell lines. This interaction appears to be mediated by the podoplanin TM domain with a requirement of the CT tail to stabilize it and is negatively modulated by glycosylation of Ser and Thr residues located between the PLAG3 and PLAG4 motifs and downstream PLAG4 in the EC domain. These findings are important to better understand the biologic function of the interplay between podoplanin and CD44, given the involvement of both glycoproteins in critical pathophysiological processes, such as the immune response, inflammation and cancer invasion and metastasis. Further studies remain to be conducted to assess the functional significance of podoplanin–CD44 interaction.

## Figures and Tables

**Figure 1 cells-09-02200-f001:**
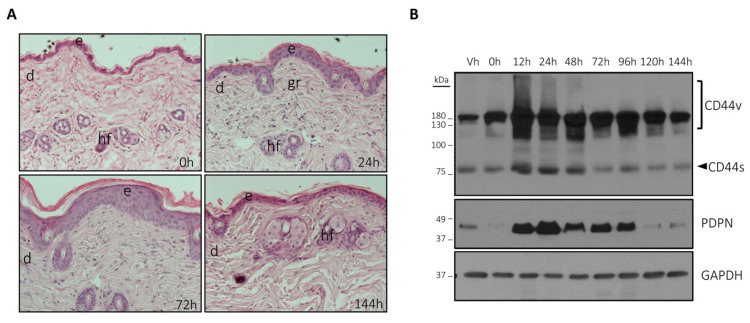
12-*O*-tetradecanoylphorbol-13-acetate (TPA) induces coordinately podoplanin, CD44 standard (CD44s) and CD44 variant (CD44v) expression in mouse-skin. (**A**) Hematoxylin–eosin staining of skin sections at the indicated times after treatment with TPA; e—epidermis; d—dermis; hf—hair follicle; gr—granulation tissue; (**B**) Western-blot analysis of podoplanin (PDPN), CD44v and CD44s in whole skin at the indicated times after treatment with TPA. Treatment with vehicle (acetone) was also included as a control. Note that vehicle slightly induced podoplanin expression, likely as a result of a cold shock. GAPDH used as a control of protein loading. Results represent three independent experiments.

**Figure 2 cells-09-02200-f002:**
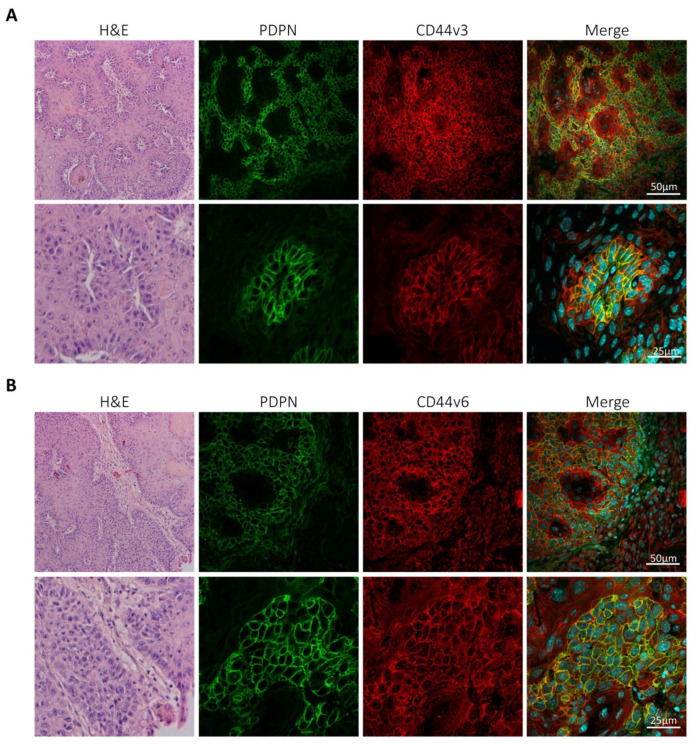
Colocalization of podoplanin with CD44v isoforms in chemically induced mouse-skin squamous carcinoma cell (SCC). Tumor skin sections were double stained with a rat mAb specific for podoplanin (PDPN) and rabbit polyclonal Abs recognizing sequences encoded by the (**A**) CD44 v3 or (**B**) v6 exon and appropriate secondary Abs. Nuclei stained with 4′,6-diamino-2-phenilindole (DAPI). Merge images show a colocalization of podoplanin and CD44 isoforms containing v3 or v6 exons in the tumors. Representative images from three different tumors in each case are shown.

**Figure 3 cells-09-02200-f003:**
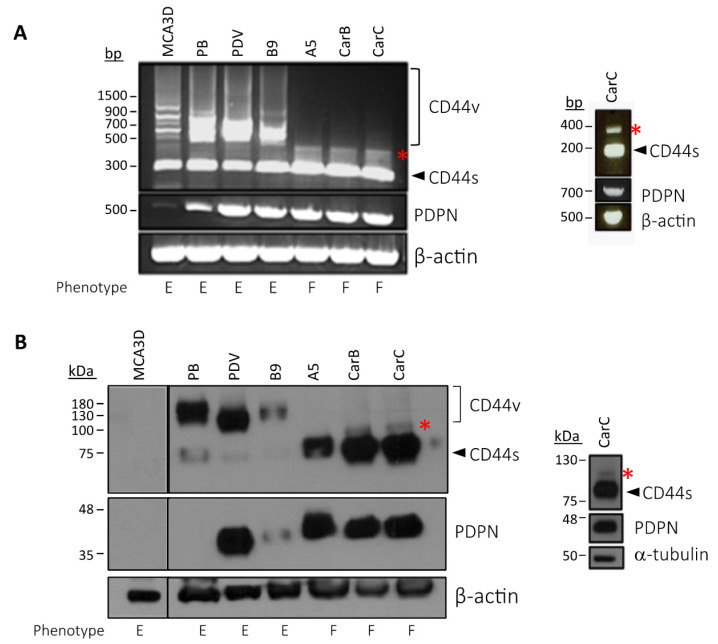
Podoplanin is co-expressed with CD44v and CD44s in transformed mouse epidermal cell lines. CD44v10 is specifically expressed in spindle carcinoma cells. (**A**) RT–PCR analysis of podoplanin and CD44 mRNA expression in the cell lines. The asterisk indicates a transcript of ~380 bp specifically expressed in differentiated spindle cell carcinomas (SpCCs) cells that after isolation (right panel) and sequencing was identified as CD44v10. β-actin was used as a control of RNA loading; (**B**) Western-blot analysis of podoplanin and CD44 protein expression in the cell lines. Asterisk indicates a protein band putatively corresponding to CD44v10. β-actin and α-tubulin (blot on the right) were used as controls of protein loading. Letters below the blots indicate the phenotypes of the cell lines: E—epithelial; F—fibroblastic. Results represent three independent experiments.

**Figure 4 cells-09-02200-f004:**
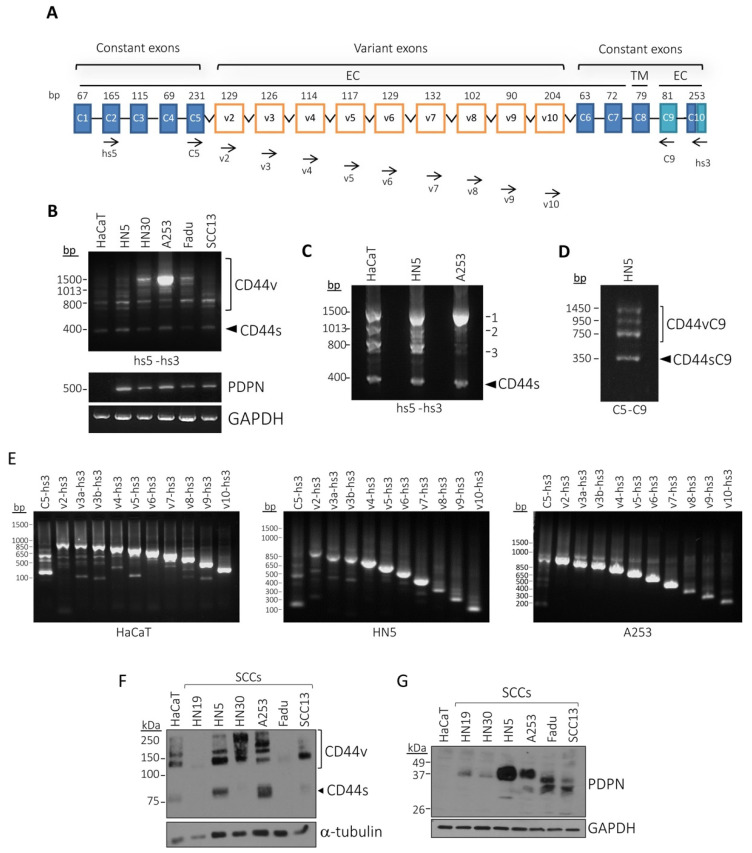
Podoplanin is co-expressed with CD44v and CD44s isoforms in human SCC cell lines. (**A**) Schematic representation of the human CD44 gene indicating the positions of the primers used in the RT–PCR analysis. Primer positions are indicated by arrowheads. Numbers above constant (C) and variant (v) exons indicate their size in base pairs. Exon v1 is not represented since this sequence is not expressed in human cells [47]. EC—extracellular domain; TM—transmembrane domain; CT—cytosolic domain; (**B**) RT–PCR analysis of podoplanin and CD44 expression in the indicated cell lines. The pair of primers hs5-hs3 was used for CD44. GAPDH was used as a control of RNA loading; (**C**) RT–PCR analysis of CD44 expression in the indicated cell lines using primers hs5-hs3. Three main bands of ~1500, ~1013 and ~800 bp, denoted as 1, 2 and 3, respectively were observed; (**D**) RT–PCR analysis of CD44 expression in HN5 cells using primers C5–C9; (**E**) exon-specific RT–PCR analysis of CD44 expression in the indicated cell lines. v2–v10, 5′ primers were used in combination with 3′ primer hs3. Exon v3 is covered by two primers (v3a and v3b) because of the existence of an alternative splice acceptor site in this exon [48]; (**F**) Western-blot analysis of CD44 protein expression. α-tubulin was used as a control of protein loading; (**G**) Western-blot analysis of podoplanin protein expression. GAPDH was used as a control of protein loading. Data represent three independent experiments.

**Figure 5 cells-09-02200-f005:**
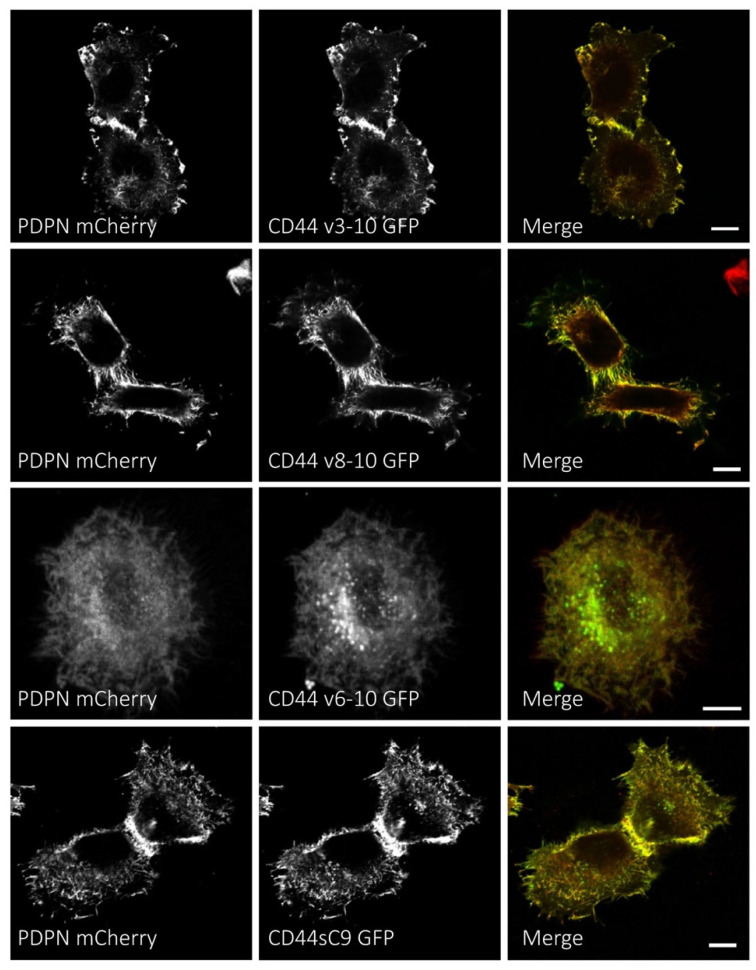
Colocalization of podoplanin with CD44v and CD44sC9 isoforms in SCC cells. HN5 cells were co-transfected with PDPN-mCherry and CD44v3-10-eGFP, CD44v8-10-eGFP, CD44v6-10-eGFP or CD44sC9-eGFP constructs and the subcellular distribution of the proteins studied by immunofluorescence and confocal microscopy. Grayscale images show the specific localization of each single protein while merged images are the result of GFP (green) and mCherry (red) combined images. Data represent 10–12 cells analyzed per condition from three independent experiments. Scalebar, 10 μm.

**Figure 6 cells-09-02200-f006:**
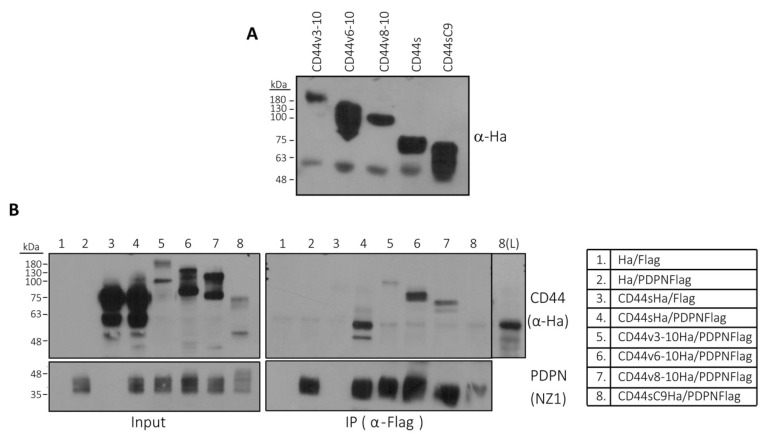
Podoplanin binds both CD44s and CD44v isoforms. (**A**) Western-blot analysis of constructs encoding the indicated CD44v and CD44s isoforms tagged with Ha after expression in HEK293T cells; (**B**) co-immunoprecipitation analysis of podoplanin and CD44 isoforms. HEK293T cells were co-transfected with FLAG-tagged human podoplanin and Ha-tagged human CD44v and CD44s isoforms. Lysates were immunoprecipitated with an ant-FLAG Ab and immunoprecipitates were run by SDS-PAGE and immunoblotted with an anti-Ha Ab (right panel). Lane 8L shows a longer exposure of CD44sC9 co-immunoprecipitate. Left panel shows the expression levels of the different CD44 constructs after transfection (input). Presence of podoplanin in the cell lysates and precipitates determined with an anti-podoplanin Ab (NZ1). Note CD44 proteins co-precipitated with podoplanin had lower sizes than fully glycosylated mature forms. Results represent three independent experiments.

**Figure 7 cells-09-02200-f007:**
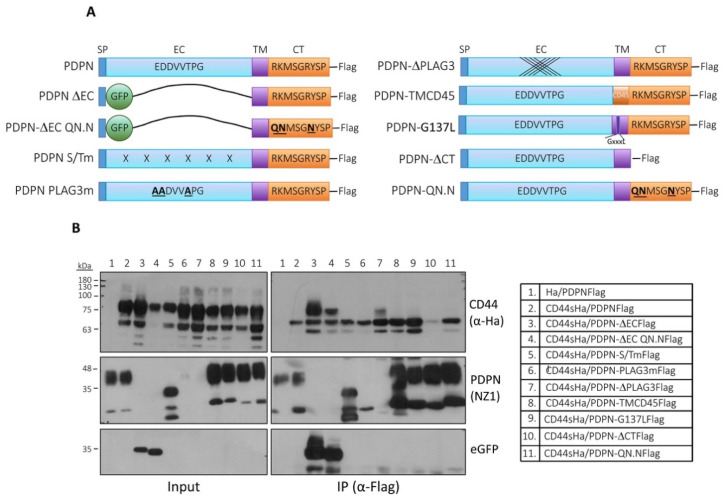
Co-immunoprecipitation of podoplanin mutants with CD44s. (**A**) Schematic representation of podoplanin wild-type and mutant constructs used for co-immunoprecipitation assays. SP—signal peptide; EC—extracellular domain; TM—transmembrane domain; CT—cytosolic domain; ΔEC—the podoplanin EC domain was substituted by GFP; QN.N—positive charged residues (RK.R) in podoplanin juxtamembrane CT domain were substituted by uncharged polar amino acids (QN.N) to impede binding to ERM proteins; S/Tm—potential O-glycosylation Ser and Thr EC residues were stochastically mutated to Ala; PLAG3m—several residues, including Thr52, whose glycosylation was found to be crucial for podoplanin interaction with CLEC-2 were mutated to Ala; ΔPLAG3—the whole PLAG3 motif was deleted; TMCD45—the podoplanin TM region was substituted by that of CD45; G137L—the GXXXG motif (GIIVG) within the TM region involved in podoplanin clustering and association with lipid rafts was mutated to GXXXL; ΔCT—the whole podoplanin CT domain was deleted; (**B**) co-immunoprecipitation analysis of podoplanin wild-type and mutant forms with CD44s. HEK293T cells were co-transfected with FLAG-tagged human podoplanin constructs and Ha-tagged human CD44s. Lysates were immunoprecipitated with an ant-FLAG Ab, and immunoprecipitates were run by SDS-PAGE and immunoblotted with an anti-Ha Ab (right panel above). Left panel shows the expression levels of the different podoplanin constructs and CD44 after transfection (input). Presence of podoplanin in the cell lysates and precipitates was determined with an anti-podoplanin mAb (NZ1). Note that this mAb does not recognize the podoplanin constructs in which the PLAG3 domain is mutated (PLAG3m or ΔPLAG3) as previously reported [51]. Accordingly, the ΔEC mutants are neither detected by the NZ1 mAb but can be recognized by an anti-GFP Ab instead. Results represent three to four independent experiments.

**Figure 8 cells-09-02200-f008:**
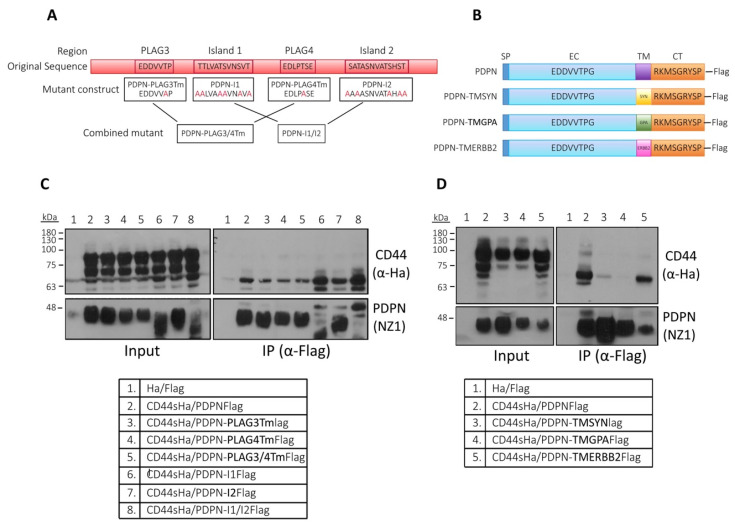
Co-immunoprecipitation of podoplanin glycosylation and transmembrane mutants with CD44s. (**A**) Schematic representation of podoplanin extracellular (EC) domain showing the amino acid sequences of four regions found to be glycosylated. Ser and Thr residues within these regions were mutated to Ala to generate the indicated podoplanin mutant constructs; (**B**) schematic representation of podoplanin wild-type and TM mutants. The podoplanin TM region was substituted by those of synaptobrevin (SYN), glycophorin A (GPA) and ERBB2 to generate the indicated chimeric mutants; (**C**,**D**) co-immunoprecipitation analysis of CD44s and (**C**) podoplanin glycosylation and (**D**) chimeric TM mutant constructs. HEK293T cells co-transfected with FLAG-tagged human podoplanin constructs and Ha-tagged human CD44s. Lysates were immunoprecipitated with an ant-FLAG Ab and immunoprecipitates were run by SDS-PAGE and immunoblotted with an anti-Ha Ab (right panels above). Left panels show the expression levels of the different podoplanin constructs and CD44s after transfection (input). The presence of podoplanin in the cell lysates and precipitates was determined with an anti-podoplanin mAb (NZ1). Results represent three to four independent experiments.

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
