# Peer review of "Interplay between Podoplanin, CD44s and CD44v in Squamous Carcinoma Cells"

_cells, 2020, doi:10.3390/cells9102200_

Round 1

Reviewer 1 Report

This manuscript describes the role of poroplanin in SCC in vitro and in vivo and they especially focused on the relationship with CD44s and CD44v. Podoplanin is one of the essential protein which could be inevitable not only for cardio-vascularization but also for cancer development. I think this manuscript will be suitable for publication after minor modifications and will cause big interests to readers of “Cells”.

  1. Figure 2

Immunofluorescence showed co-localization of podoplanin and CD44v isoforms containing v3 and v6, respectively. These clear images are good enough, however, the merged image always show yellow right. What are light-blue spots on the Merged images?

  1. Figure 3

Another immunofluorescence showed co-localization of podoplanin (mCherry) and four isoforms of CD44v, and the merge clearly showed yellowish color for merged area. However, mCherry always shows red color, and GFP always shows green color, then the merged image shows yellow color. These multi-panel figure will make readers confused why the merge of white and white could be yellow?

  1. The allocation of figures should be reconsidered for easier reading. I strongly recommend two points as follows:
  • Figure 6 should be allocated at the end of 3.5., not at the end of 3.6.
  • Page 14 and 15 should be swapped.

Author Response

"This manuscript describes the role of poroplanin in SCC in vitro and in vivo and they especially focused on the relationship with CD44s and CD44v. Podoplanin is one of the essential protein which could be inevitable not only for cardio-vascularization but also for cancer development. I think this manuscript will be suitable for publication after minor modifications and will cause big interests to readers of “Cells”."

1.Figure 2
"Immunofluorescence showed co-localization of podoplanin and CD44v isoforms containing v3 and v6, respectively. These clear images are good enough, however, the merged image always show yellow right. What are light-blue spots on the Merged images?"

Author reply:We are sorry for this confusion as we did not include the required information to understand this in the figure legend. The blue spots on the merged images are nuclei stained with DAPI. We have forgotten to mention it in the Figure legend or the Materials and Methods section. These gaps have been corrected in the revised version: page 5, lines 182-183; and page 7, lines 276-277.

2. Figure 3
"Another immunofluorescence showed co-localization of podoplanin (mCherry) and four isoforms of CD44v, and the merge clearly showed yellowish color for merged area. However, mCherry always shows red color, and GFP always shows green color, then the merged image shows yellow color. These multi-panel figure will make readers confused why the merge of white and white could be yellow?"

Author reply:We believe the reviewer refers to Figure 5, since Figure 3 does not show any immunofluorescence images. In this Figure, immunofluorescence localization of single proteins is shown in Black & White while only merged images are shown in RGB color. Using Black & White images is a common practice in many labs working with immunofluorescence techniques in the cell biology field. The reason for this fact is that, grayscale images (or green) are always easier to see, while blue and red, are normally not as easy to spot. This usually helps to show a precise subcellular localization of a single protein. Although this is a common practice in many cell biology articles, we agree with the reviewer that this fact could be confusing for some readers. For this reason, to make this point clearer we have included a sentence regarding this point in the figure legend (pag. 11, lines 465-466).

"3. The allocation of figures should be reconsidered for easier reading. I strongly recommend two points as follows:
• Figure 6 should be allocated at the end of 3.5., not at the end of 3.6.
• Page 14 and 15 should be swapped."

Author reply:The reviewer is right in that this Figure is better allocated at the end of 3.5. However, in doing so, the Figure itself and the legend will be split off in different pages. For this reason, we allocated the whole Figure below 3.6.
On the other hand, pages 14 and 15 have been swapped in the revised version, as recommended by the reviewer.

Reviewer 2 Report

In this manuscript Montero-Montero et al describe experiments investigating the interplay between PDPN and CD44 in squamous carcinoma cells. This is an impressive study involving a variety of cell and molecular experiments. Results from these analyses indicate that TPA increases the production of PDPN and CD44s and CD44v isoforms related to skin and squamous cell carcinoma progression. In addition, they indicate that PDPN and CD44 interact with each other through the PDPN transmembrane domain with direction from its cytoplasmic domain. This relationship is likely to direct stemness, polarity, and cell motility. 

Overall the work is presented well, with clear data and logical interpretations. These data constitute a significant contribution to the field. This work is impressive and is likely to be very much appreciated by the PDPN and general research communities.

Only minor concerns stand out related to English grammar. For example, the title “Interplay of podoplanin” should likely read “Interplay between podoplanin”. Such concerns could likely be addressed by a native English speaker or editorial staff.

Author Response

Reviewer 2
"In this manuscript Montero-Montero et al describe experiments investigating the interplay between PDPN and CD44 in squamous carcinoma cells. This is an impressive study involving a variety of cell and molecular experiments. Results from these analyses indicate that TPA increases the production of PDPN and CD44s and CD44v isoforms related to skin and squamous cell carcinoma progression. In addition, they indicate that PDPN and CD44 interact with each other through the PDPN transmembrane domain with direction from its cytoplasmic domain. This relationship is likely to direct stemness, polarity, and cell motility.
Overall the work is presented well, with clear data and logical interpretations. These data constitute a significant contribution to the field. This work is impressive and is likely to be very much appreciated by the PDPN and general research communities.
Only minor concerns stand out related to English grammar. For example, the title “Interplay of podoplanin” should likely read “Interplay between podoplanin”. Such concerns could likely be addressed by a native English speaker or editorial staff."
We thank the kind comments of the reviewer on the article. We have revised the whole text and corrected some errors and errata. As suggested by the reviewer, we have substituted “Interplay of podoplanin…” by “Interplay between podoplanin…” in the title.

Thank you for your consideration.
Sincerely,
Ester Martín-Villar